# Targeted memory reactivation in human REM sleep elicits detectable reactivation

**Mahmoud EA Abdellahi[1], Anne CM Koopman[1], Matthias S Treder[2], Penelope A Lewis[1]\***

[1]School of Psychology, Cardiff University Brain Research Imaging Centre (CUBRIC), Cardiff, United Kingdom; [2]School of Computer Science and Informatics, Cardiff University, Cardiff, United Kingdom

**Abstract** It is now well established that memories can reactivate during non-rapid eye movement (non-REM) sleep, but the question of whether equivalent reactivation can be detected in rapid eye movement (REM) sleep is hotly debated. To examine this, we used a technique called targeted memory reactivation (TMR) in which sounds are paired with learned material in wake, and then re-presented in subsequent sleep, in this case REM, to trigger reactivation. We then used machine learning classifiers to identify reactivation of task-related motor imagery from wake in REM sleep. Interestingly, the strength of measured reactivation positively predicted overnight performance improvement. These findings provide the first evidence for memory reactivation in human REM sleep after TMR that is directly related to brain activity during wakeful task performance.

## Editor's evaluation

This valuable work in human subjects reports that sounds that were associated with specific memories during waking behaviors can trigger the reactivation of these memory representations during REM sleep. Convincing evidence is provided to support the conclusions. The work expands our understanding of memory processing during sleep.

**\*For correspondence:**
lewisp8@cardiff.ac.uk

**Competing interest:** The authors declare that no competing interests exist.

## Introduction

The reactivation of memories in non-rapid eye movement (non-REM) sleep is widely supported by evidence from numerous species (*Rasch and Born, 2013*; *Bendor and Wilson, 2012*; *Wang et al., 2019*; *Ji and Wilson, 2007*), but it is still unclear whether equivalent reactivation occurs in rapid eye movement (REM) sleep. The first evidence for non-REM reactivation came from rodents (*Pavlides and Winson, 1989*; *Wilson and McNaughton, 1994*), and such reactivation has subsequently been identified in humans using EEG classifiers (*Wang et al., 2019*; *Belal et al., 2018*; *Schreiner et al., 2018*; *Cairney et al., 2018*; *Abdellahi et al., 2023*), fMRI (*Shanahan et al., 2018*; *Rasch et al., 2007*), and intracranial recordings (*Zhang et al., 2018*). Only a few rodent studies have shown evidence for reactivation in REM sleep (*Pavlides and Winson, 1989*; *Louie and Wilson, 2001*; *Hennevin et al., 1995*; *Poe et al., 2000*), but support for such reactivation comes from human work by *Schönauer et al., 2017*, who used EEG classifiers to elegantly distinguish between the REM sleep on nights after training on two very different tasks. Nevertheless, the reinstatement of EEG activity in REM that directly relates to EEG activity during a task in wake has yet to be demonstrated in humans. Targeted memory reactivation (TMR), a technique which allows the active triggering of memory reactivation, is linked to both neural (*Lewis and Bendor, 2019*; *Berkers et al., 2018*) and behavioural (*Hu et al., 2019*; *Pereira et al., 2022*; *Oudiette and Paller, 2013*) plasticity when applied in non-REM sleep.

**eLife digest** Sleep is crucial for rest and recovery, but it also allows the brain to process things it has learned while awake. This is why a person may go to bed frustrated with learning a tune on the piano but wake up the next morning ready to play it without fumbling. For this to happen, it is thought that memories must be reactivated during sleep – something which can be studied by monitoring brain activity. While it has been shown that memory reactivation occurs in some stages of human sleep, it was unclear whether it occurred in a specific stage known as REM sleep – which is important for learning.

To study memory reactivation during REM sleep, Abdellahi et al. recruited volunteers and monitored their brain activity during an 'adaptation night' when certain sounds played as they slept. The following day, memories – such as an image or pressing a certain button – were paired with the sounds, which were replayed during REM sleep the following night to trigger memory reactivation (experimental night). Abdellahi et al. measured how strongly brain activity during each night related to the waking activity when the sound pairing tasks were imagined and compared the adaptation and experimental nights. The experimental night showed clear signs of memory reactivation after the sounds were played during REM sleep, suggesting that the sounds triggered memories of the associated images or buttons.

These findings show that in humans, brain activity patterns that indicate memory reactivation can be identified during REM sleep. The work paves the way for future studies into the characteristics of this memory reactivation and how to trigger it in a way that leads to improvements in memory.

TMR provides a temporal window for reactivation to occur as a result of the external cue, and this can be used to evaluate the detectability of such reactivation. In the current study, we used TMR to build on the work of *Schönauer et al., 2017*, by looking for task-related activity after TMR cues during REM sleep. We thus searched for a direct link between brain activity associated with a memory in wake and cued reactivation in REM sleep. Given that theta activity predominates during REM (*Lomas et al., 2015*; *Nishida et al., 2009*), we also wanted to gain a better understanding of the theta response after TMR cues in REM, and how this relates to reactivation.

We chose the serial reaction time task (SRTT), which is known to be sleep sensitive (*Spencer et al., 2006*; *Born and Wilhelm, 2012*), to examine these questions. The SRTT is modulated by TMR in non-REM sleep (*Cousins et al., 2014*; *Cousins et al., 2016*; *Rakowska et al., 2021*) but has also been strongly linked to REM (*Maquet et al., 2000*; *Peigneux et al., 2003*; *Laureys et al., 2001*). For instance, brain areas which were activated during the execution of an SRTT have been shown to reactivate during subsequent REM (*Maquet et al., 2000*; *Peigneux et al., 2003*). Furthermore, greater connectivity between premotor cortex, posterior parietal cortex, and bilateral pre-supplementary motor areas has been observed during REM after training on this task (*Laureys et al., 2001*). These studies build on the literature suggesting that memories may reactivate in REM to suggest that the SRTT specifically can be reprocessed in this sleep stage.

In our SRTT, participants were presented with audio-visual cues and responded by pressing four buttons (two with each hand), visual cues are shown in *Figure 1—figure supplement 1*. These cues were organised into an implicit 12-item sequence which was practiced repeatedly. The cue-sounds were then replayed during subsequent REM sleep to trigger the associated memories of left- and right-hand presses (*Figure 1c*). We trained participants on two sequences and replayed only one of them in sleep. As a control, we included an adaptation night in which participants slept in the lab and we played the same cue-sounds that would later be played during the experimental night. This provided an important control, as a null finding from this adaptation night would ensure that we are decoding actual memories, not just sounds. We used these data to discriminate between neural reactivation of left- and right-hand button presses using linear classification on EEGs during REM sleep, see Methods for more details.

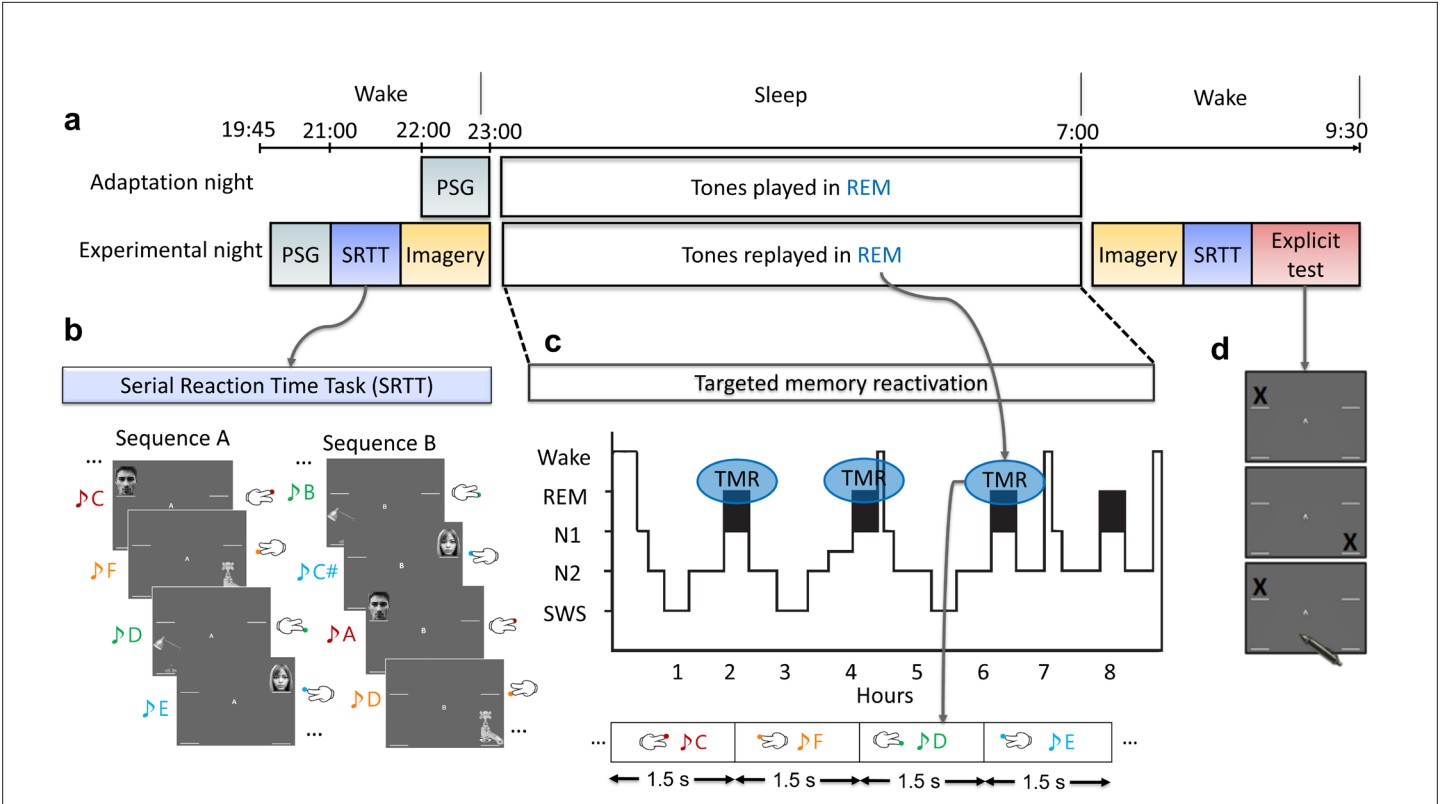

**Figure 1.** Experimental design. (**a**) The experiment consisted of two nights: an adaptation and an experimental night. On both nights, participants were wired up for polysomnography (PSG) and we recorded brain activity throughout the night. In both nights, tones were presented during rapid eye movement (REM) sleep. However, prior to the experimental night, participants completed training on the serial reaction time task (SRTT), and then performed an imagery task in which they were cued with pictures and sounds, but only imagine performing the finger tapping (without movement). PSG was recorded throughout these tasks. After waking up, from the experimental night, participants completed the motor imagery and the SRTT again, and finally did the explicit recall task. (**b**) In the SRTT, images were presented in two different sequences each with a different set of tones. Each image was associated with a unique tone and required a specific button press. In the imagery task, participants heard the tones and saw the images as in the SRTT, but only imagined pressing the buttons. This imagery data was used for classification, as it has cleaner signals compared to SRTT since there are no movement artefacts. (**c**) The sounds of only one learned sequence (cued sequence) were played during REM sleep to trigger the associated memories of left- and right-hand presses. (**d**) Participants were asked to mark the order of each sequence on paper as accurately as they could remember during the explicit recall test after sleep.

The online version of this article includes the following figure supplement(s) for figure 1:

**Figure supplement 1.** Illustration of the four images that appeared in the task: two faces and two objects.

## Results

### Elicited response pattern after TMR cues

We looked at the TMR-elicited response in both time-frequency and ERP analyses using a method similar to the one used in *Cairney et al., 2018*, see Methods. As shown in *Figure 2a*, the EEG response showed a rapid increase in theta band followed by an increase in beta band starting about 1 s after TMR onset. REM sleep is dominated by theta activity, which is thought to support the consolidation process (*Diekelmann and Born, 2010*), and increased theta power has previously been shown to occur after successful cueing during sleep (*Schreiner and Rasch, 2015*). We therefore analysed the TMR-elicited theta in more detail. Focussing on the first second post-TMR onset, we found that theta was significantly higher here than in the baseline period, prior to the cue [-300 -100]ms, for both adaptation (Wilcoxon signed rank test, n=14, p<0.001) and experimental nights (Wilcoxon signed rank test, n=14, p<0.001). The absence of any difference in theta power between experimental and adaptation conditions (Wilcoxon signed rank test, n=14, p=0.68) suggests that this response is related to processing of the sound cue itself, not to memory reactivation. Turning to the ERP analysis, we found a small increase in ERP amplitude immediately after TMR onset, followed by a decrease in amplitude

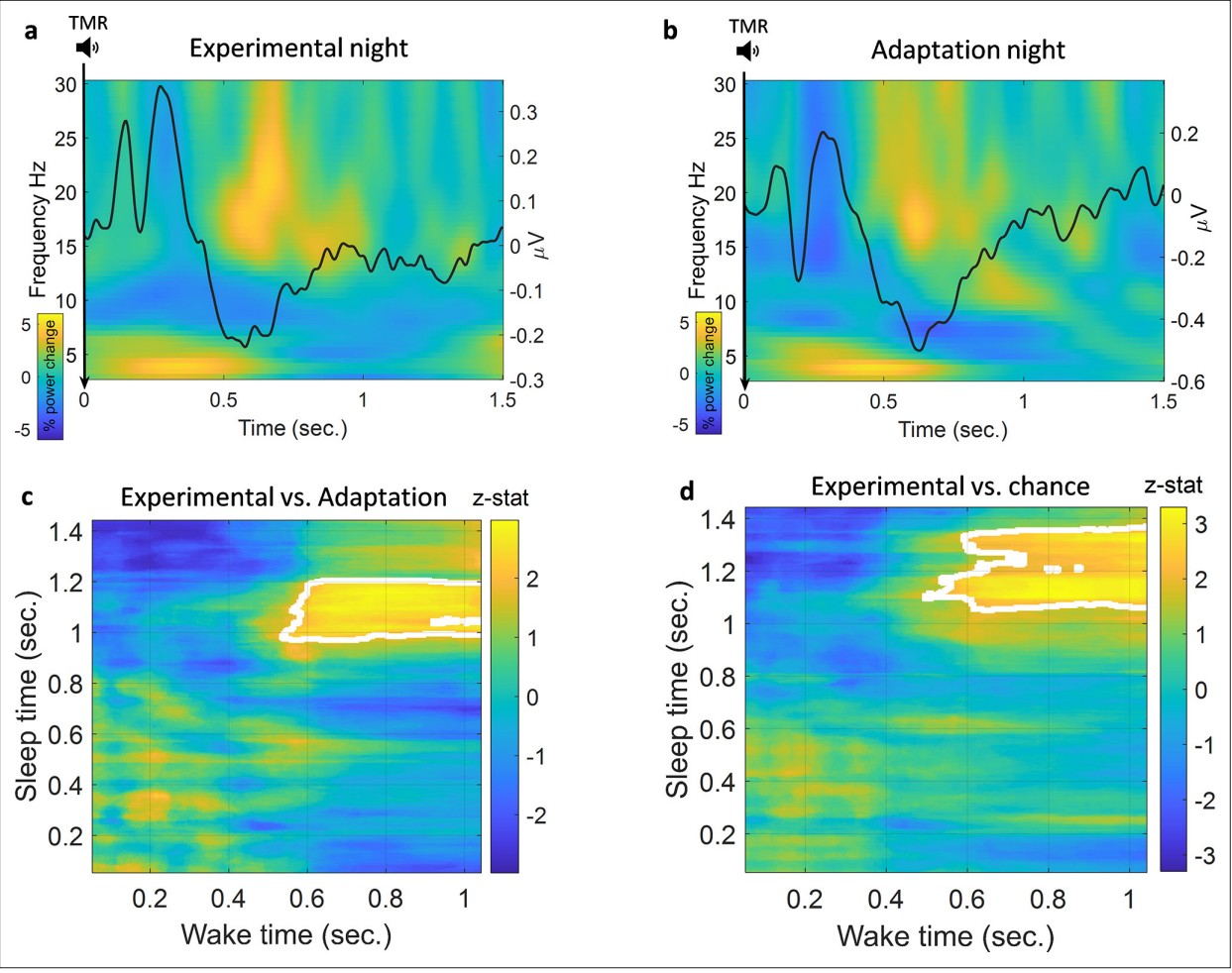

**Figure 2.** Classification of left vs. right hand. (**a**) Time-frequency and ERP analyses of the experimental night. Power percentage changes from the baseline period are shown with colours. The solid black line represents the average results of all ERP analyses from all participants. (**b**) Time-frequency analysis and ERP analysis of the adaptation night. (**c**) Comparison between classification performance on experimental and adaptation nights reveals a significant effect described by a cluster which shows a higher classification performance for the experimental night compared to adaptation (n=14, p=0.004), a z-statistic value at every point is shown and cluster edges are marked with white after correcting for multiple comparisons with cluster-based permutation (see Methods for details). (**d**) Classification performance for the experimental night was also significantly higher than chance (AUC: 0.5) as shown by the cluster after correction (n=14, p=0.009).

The online version of this article includes the following figure supplement(s) for figure 2:

**Figure supplement 1.** Classification of left vs. right hand.

**Figure supplement 2.** Classification of left vs. right hand when theta band [4 8]Hz is filtered out.

**Figure supplement 3.** Training with wake vs. training with sleep.

**Figure supplement 4.** Searchlight analysis with linear classification on motor imagery data.

**Figure supplement 5.** Example of the time × time classification procedure wherein one time point is used from sleep to build a classifier model and all wake time points were used for testing.

500 ms after the cue. Comparison of ERPs from experimental and adaptation nights showed no significant difference, (n=14, p>0.1). Similar to the time-frequency result, this suggests that the ERPs observed here relate to the processing of the sound cues rather than any associated memory.

## Detection of memory reactivation after TMR cues

We trained our EEG classification models using sleep data and then tested them on wake data. We chose to do this because training a classification model on wake could cause the model to weigh features which are dominant in wake very highly even if those features were absent from sleep. By

training our models on sleep data, we ensured that the features associated with reactivation were used by the models, and the models were thus able to look for these in the stronger, less noisy, signals recorded during wake. We used a searchlight approach to locate the channels of interest, see Methods for details. We then used a linear discriminant analysis (LDA) classifier in a time × time classification procedure (*King and Dehaene, 2014*). We also repeated the classification process using the adaptation night to be certain that the classification was not caused by sound-induced effect or EEG noise rather than reactivation of the encoded memory. Finally, we compared the results from the two nights, both to each other and to chance level using cluster-based permutation, see Methods. Comparison of the experimental night to the adaptation night showed a significantly higher area under the receiver operating characteristic curve (AUC) for the experimental night, described by a cluster (n=14, p=0.004) around 1 s after the onset of the cue at time 0, *Figure 2c*. In the adaptation night, no significant classification strength was found compared to chance (AUC=0.5), demonstrating that classification of this control condition did not differ from chance level. By contrast, comparison of the experimental night against chance showed a significantly higher AUC around 1 s after the onset of the cue for the experimental night, (n=14, p=0.009), *Figure 2d*. Time × time classification AUC plots are shown in *Figure 2—figure supplement 1*. These findings are significant when evaluated against both control and chance level, clearly demonstrating that we can detect memory reactivation after TMR cues in REM sleep.

As described in the section above, TMR elicited an increase in theta power in both experimental and adaptation nights. Because theta responses after TMR have been implicated in reactivation and consolidation (*Schreiner and Rasch, 2017*), we wanted to determine whether this increased theta power could be the carrier of reactivation and thus hold the discriminating features of classes. We therefore band-pass filtered our data and re-ran our classification analysis using only theta band. Interestingly, classification did not differ from chance in this theta-only analysis (p>0.3), suggesting that theta activity itself does not hold the information used to detect reactivation. For completeness, we also conducted a positive control in which we filtered out theta band and performed our classification analysis. We found a pattern of significant difference similar to that of the original classification in *Figure 2c and d*. Specifically, there was a significant difference against both chance level (n=14, p=0.007, corrected with cluster-based permutation) and the adaptation night (n=14, p=0.009, corrected with cluster-based permutation), as shown in *Figure 2—figure supplement 2*.

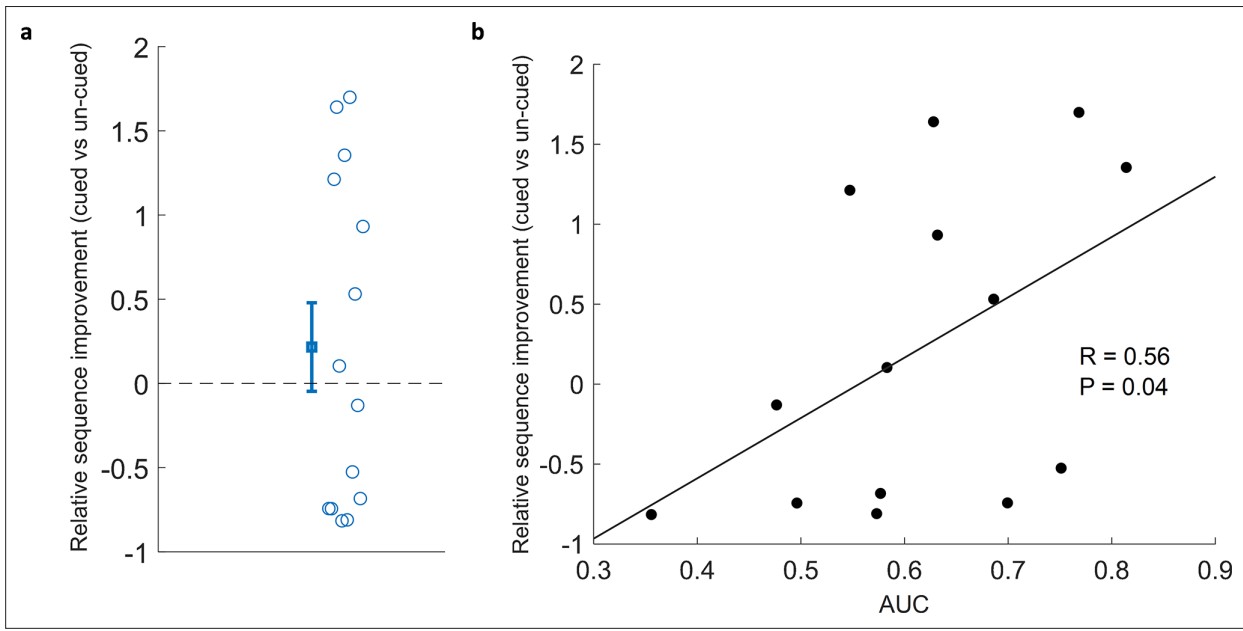

**Figure 3.** Correlation of classification with cued sequence improvement. (**a**) Relative sequence improvement (cued vs un-cued) did not show significant improvement (Wilcoxon signed rank test, n=14, p=0.43). (**b**) Classification performance positively correlated with relative sequence improvement (Spearman correlation, n=14, r=0.56, p=0.04).

## Correlation of classification performance with behaviour

Because TMR in SWS leads to reliable behavioural benefits in the SRTT *Cousins et al., 2014*; *Cousins et al., 2016*, we were interested to know whether applying this same manipulation in REM sleep lead to any measurable advantage. Our examination of overnight change showed that relative sequence improvement (cued vs un-cued) did not differ from chance (Wilcoxon signed rank test, n=14, p=0.43), *Figure 3a*, see Methods for details. This null result regarding the impact of TMR mirrors the finding of *Cousins et al., 2016*, where there was no significant result of stimulation, but there was a correlation with sleep spindles, which are thought to be markers of reactivation. We have a much stronger proxy for reactivation in this study in the form of our classification measure. We therefore set out to determine whether there was a relationship between behavioural improvement and the extent to which we could classify reactivation. This revealed a positive correlation (n=14, r=0.56, p=0.04), *Figure 3b*.

Finally, we wanted to know whether the extent to which participants learned the sequence during training might predict the extent to which we could identify reactivation during subsequent sleep. We therefore checked for a correlation between classification performance and pre-sleep performance to determine whether the degree of pre-sleep learning predicted the extent of reactivation, this showed no significant correlation (n=14, r=−0.39, p=0.17). Together, these findings suggest that while our TMR cueing did not lead to an overall significant benefit in behavioural performance, the extent to which we can detect reactivation in REM sleep did predict the extent to which sequence memory improves overnight, and this was not dependent on pre-sleep learning.

## Discussion

In this paper we demonstrated that memory reactivation after TMR cues in human REM sleep can be decoded using EEG classifiers. Such reactivation appears to be most prominent about 1 second after the sound cue onset. Interestingly, although TMR also elicited an increase in theta power, our data suggest that this response is related to the tone, and does not embody the reactivation in and of itself.

### Temporal structure of reactivation and TMR-elicited theta activity

Our data suggest that detectable memory reactivation is temporally consistent among trials and participants, as reflected in the timing of the clusters in *Figure 2c and d*. During wake, classification is more prominent around 0.6 s after the cue, whereas in REM it starts about 0.4 s later. This delayed start could potentially be due to the brain taking more time to process information and reactivate a memory during REM sleep than during wake. In fact, reactivation in SWS has also been shown to be delayed, appearing approximately from 1 s to 4.5 s after cue onset in a task of associating spatial locations with left- and right-hand movements (*Wang et al., 2019*), around 2 s after cue onset in a picture memory task (*Cairney et al., 2018*), and up to 10 s after cue onset in rodents (*Bendor and Wilson, 2012*).

REM sleep is dominated by theta activity, which is linked to attention during wake (*Biel et al., 2021*) and is more prominent with higher executive control (*Magosso et al., 2021*). Wakeful theta is also associated with the encoding of new information and memory processing (*Vertes, 2005*). Furthermore, studies in non-REM have shown that TMR triggers an increase in theta followed by an increase in sigma band (*Cairney et al., 2018*; *Schechtman et al., 2021*). Our data are in keeping with this, as we show that TMR elicited a power increase in theta band in REM sleep irrespective of whether the cues are meaningful (experimental night) or not (control night). We also demonstrate that theta does not embody the reactivation we are detecting, since we cannot classify reactivation using only this frequency. We therefore conclude that the theta activity we observed following TMR cues was related to the processing of the sound cues and not a representative of reactivation.

### Correlation with behaviour

The performance of our REM classifiers positively predicts overnight improvement on sequence memory. Similar correlations between classification performance and behaviour have been found in non-REM TMR (*Wang et al., 2019*; *Shanahan et al., 2018*; *Zhang et al., 2018*). This could lead to the speculation that more reactivation means more consolidation, and therefore better post-sleep performance. However, TMR in REM sleep lead to no overall benefit in performance in either our current dataset (*Koopman et al., 2020*) or previous studies (*Rasch et al., 2007*; *Cordi et al., 2014*).

Furthermore, examination of natural (un-cued) sleep showed a correlation between overnight memory improvement and the extent of memory-linked activity identified in non-REM but not REM sleep (*Schönauer et al., 2017*), potentially suggesting a difference in the function of replay in these two sleep stages. We are at a very early phase in understanding what TMR does in REM sleep, however we do know that the connection between hippocampus and neocortex is inhibited by the high levels of acetylcholine that are present in REM (*Hasselmo, 1999*). This means that the reactivation which we observe in the cortex is unlikely to be linked to corresponding hippocampal reactivation, so any consolidation which occurs as a result of this is also unlikely to be linked to the hippocampus. The SRTT is a sequencing task which relies heavily on the hippocampus, and our primary behavioural measure (sequence-specific skill [SSS]) specifically examines the sequencing element of the task. Our own neuroimaging work has shown that TMR in non-REM sleep leads to extensive plasticity in the medial temporal lobe (*Cousins et al., 2016*). However, if TMR in REM sleep has no impact on the hippocampus then it is quite possible that it elicits cortical reactivation and leads to cortical plasticity but provides no measurable benefit to SSS. Alternatively, because we only measured behavioural improvement right after sleep it is possible that we may have missed behavioural improvements that would have emerged several days later, as we know can occur in this task (*Rakowska et al., 2021*). We should also mention our relatively low number of participants (n=14) is a limitation to this correlation and we recommend that future studies confirm the same relationship.

## Parallel characteristics of reactivations in REM and non-REM

While there is already a growing body of literature about TMR cued reactivation in non-REM sleep (*Rasch and Born, 2013*; *Bendor and Wilson, 2012*; *Wang et al., 2019*; *Cairney et al., 2018*; *Schreiner et al., 2021*), our findings provide initial information about human memory reactivation using TMR in REM. In fact, our data suggest some parallels between reactivation in these two sleep stages. For instance, similar to non-REM (*Cairney et al., 2018*), TMR-locked reactivation in REM is delayed compared to wake after cue onset. Furthermore, the reactivations we were able to detect appear to occur in roughly the same area of the cortex as is used in performing the task, and the extracted features of reactivation are similar to those extracted during wake activation which is why it is detectable using our machine learning models.

## Authenticity of detected memory reactivation

We ensured that the pattern detected by our EEG classifiers is a genuine memory reactivation using two procedures. Firstly, we use an adaptation night, during which participants heard the TMR sounds before these were paired with any memory to ensure that our classification results from the experimental night were not caused by responses to sounds alone. Secondly, in developing our classifier, we captured the neural fingerprint associated with reactivation of a memory using the EEG pattern occurring after TMR cues in sleep. We then directly related this to the pattern of brain activity during wakeful imagining of the task. This was achieved by training our classification models on sleep EEG and testing them on wakeful imagining. Together, these procedures ensured that our classification stems from memory reinstatement and is related to genuine re-processing of memories during sleep. Our work builds on other studies showing discriminability of cued categories in sleep data without the inclusion of wake (*Cairney et al., 2018*; *Schönauer et al., 2017*), as well as on an approach that includes only the features that caused category discrimination in wake (*Wang et al., 2019*). We have explicitly examined the differential impact of training our classifiers with wake and training them with sleep and shown in *Figure 2—figure supplement 3* that the classifiers trained with sleep are more robust in decoding the elicited reactivation. We therefore recommend that future studies adopt a similar approach by allowing the models to train on sleep data.

## Conclusion

The question of whether memories reactivate in REM as well as non-REM sleep has been debated for some years. REM reactivation has been suggested by modelling (*Hasselmo, 2008*) and evidence of learning-dependent activation in human REM sleep has been observed (*Maquet et al., 2000*; *Peigneux et al., 2003*). However, null behavioural findings from human REM TMR studies *Rasch and Born, 2013*; *Rasch et al., 2007* have led to scepticism amongst sleep researchers. Our current findings provide clear evidence that TMR in REM can elicit detectable reactivation. Furthermore, our analysis

uncovers several important properties of REM reactivation, showing strong parallels with reactivation in non-REM sleep. More work is needed to explore how such reactivation links to behavioural and neural plasticity, and how this differs across a variety of cognitive tasks.

## Methods

### Participants

EEG and behavioural data were collected from human participants (n=16) (8 females, 8 males, and age mean: 23.6). Two participants were excluded due to technical problems, the rest of the datasets were included in the analyses (n=14). All participants had normal or corrected-to-normal vision, normal hearing, and no history of physical, psychological, neurological, or sleep disorders. Responses in pre-screening questionnaires reported no stressful events, a regular sleep-wake rhythm in the month before the study, and no night work or cross-continental travel in the 2 months before the study. All participants reported non-familiarity with the SRTT and all of them were right-handed. Participants did not consume alcohol in the 24 hr before the study or caffeine in the 12 hr prior to the study or perform any extreme physical exercise or nap.

### Experimental design

The SRTT was adapted from *Cousins et al., 2014*, and participants performed SRTT before and after sleep. In the SRTT, sounds cued four different finger presses. During wakeful encoding, participants learned two 12-item sequences, A and B, A: 1 2 1 4 2 3 4 1 3 2 4 3 and B: 2 4 3 2 3 1 4 2 3 1 4 1. The location indicated which key on the keyboard to press as quickly and accurately as possible afterwards the next image appeared, locations and buttons associations were: 1 – top left corner = left shift key; 2 – bottom left corner = left Ctrl; 3 – top right corner = up arrow; 4 – bottom right corner = down arrow. Sequences had been matched for learning difficulty; both contained each item three times. The blocks were interleaved so that a block of the same sequence was presented no more than twice in a row, and each block contained three repetitions of a sequence. There were 24 blocks of each sequence (48 blocks in total), after each block a 15 s pause which could be extended by participants if they wish, during the pause participants were informed of their reaction time and error rate for the last block. We used two sets of pure musical tones, one consisted of low tones within the fourth octave (C/D/E/F) and the second consisted of high tones within the fifth octave (A/B/C#/D). Each sequence was paired with either low tones or high tones. These tone groups were counterbalanced across participants. The 48 blocks of sequences A and B were followed by four blocks that contained random sequences, which were used to help us isolate improvements in reaction time which were specific to sequence learning. Two of these random blocks were paired with the tone group of one sequence that was later replayed in REM sleep, and the other two were paired with the tone group of the other non-reactivated sequence (*Abdellahi et al., 2023*; *Rakowska et al., 2021*; *Koopman et al., 2020*).

On the centre of the screen 'A' and 'B' appeared while participants performed the task, and they knew that there were two sequences. However, they were not asked to explicitly learn the sequences. We counterbalanced which sequence was presented first and which sequence was reactivated across participants. At the beginning of each trial (time 0), a 200 ms tone was played, and at the same time a visual cue appeared in one of the corners of the screen. Participants were instructed to keep individual fingers of their left and right hand on the left and right response keys, respectively. Visual cues were neutral objects or faces, (*Figure 1—figure supplement 1*), used in previous study (*Cousins et al., 2014*). Visual cues appeared in the same position for each sequence (1=male face, 2=lamp, 3=female face, 4=water tap). Visual cues stayed on the screen until the correct key was pressed. After completing the SRTT, participants performed the imagery task by seeing and hearing the same cues as in the initial training task, but only imagining pressing the buttons without movement. This task consisted of 30 interleaved blocks (15 of each sequence) and was presented in the same order as during the SRTT. We used this imagery data for classification as it has higher signal-to-noise ratio since there is no movement compared to actual finger presses. Each trial consisted of a tone and a visual stimulus, the latter being shown for 270 ms and followed by an 880 ms inter-trial interval. There were no random blocks during the imagery task and no performance feedback during the pause between blocks. As a control, participants were asked to sleep in the lab before doing the SRTT training. The control night followed the same criteria as the actual experiment with the difference that the sounds

were not yet associated with any task. Participants performed the task again in the morning but this time they did first motor imagery then SRTT. They were then asked to try to remember the locations of the images of the two sequences to test their recall for each of the sequences.

## The delivery of TMR cues during REM sleep

Cues were delivered to participants during stable REM sleep which was identified using standard American Academy of Sleep Medicine (AASM) criteria (*Richard et al., 2015*). Tones were presented in the correct order for our sequence, with a pause of 1500 ms in between tones in a sequence, and a 20 s break at the end of each sequence. If participants showed any signs of arousal, or left REM sleep, cueing was paused immediately. Cueing then continued from where it had left off when they returned to stable REM sleep again.

## EEG recording and pre-processing

EEG signals were acquired from 21 electrodes, following the 10–20 system. Thirteen electrodes were placed on standard locations: FZ, CZ, PZ, F3, F4, C5, CP3, C6, CP4, P7, P8, O1, and O2. Other electrodes were: left and right EOG, three EMG electrodes on the chin, and the ground electrode was placed on the forehead. Electrodes were referenced to the mean of the left and right mastoid electrodes. The impedance was <5 kΩ for each scalp electrode, and <10 kΩ for each face electrode. Recordings were made with an Embla N7000 amplifier and RemLogic 1.1 polysomnography (PSG) software (Natus Medical Incorporated). PSG recordings were scored by trained sleep scorers and only the segments scored as REM sleep were kept for further analyses.

Data were collected at 200 Hz sampling rate. EEG signals were band-pass filtered in the frequency range from (0.1 to 50 Hz). Subsequently, trials were cleaned based on statistical measures consisting of variance and mean. Trials were segmented –0.5 s to 3 s relative to the onset of the cue. EEG traces falling two standard deviations higher than the mean were considered outliers and trials were rejected if more than 25% of the channels were outliers. If trials were bad in less than 25% of the channels, they were interpolated using triangulation of neighbouring channels. Thus, 9.8% and 10.5% of trials were considered outliers and removed from the experimental night data and the adaptation night, respectively.

Data was subsequently analysed with independent component analysis (ICA), to remove eye movement artefacts which can occur during REM. Components identified by the ICA that were significantly correlated with the signals from the eye electrodes (corrected for multiple comparisons) were removed. All channels for each participant were manually inspected and artefacts were removed.

## Time-frequency representation and ERP analysis

Our time-frequency analysis used a similar method to that used in *Cairney et al., 2018*. We used a hanning taper with 5 cycles that was convolved with the signals. We used 0.5 Hz frequency steps and 5 ms time steps. Power values are shown in the range of 4–30 Hz, *Figure 2a and b*. We also used a similar baseline of –300 ms to –100 ms pre-stimulus. The reported values represent the percentage of power change from baseline. The shown plots are the grand average from all participants and all channels. The same process was applied to both experimental and adaptation nights. For the ERP analysis, we adopted a similar approach to that in *Cairney et al., 2018*. We identified a baseline period of –200 ms to 0 ms and we reported the grand average from all participants and all channels. Small values of amplitudes are caused by the overall smoothing that would happen as a result of averaging many trials and taking the grand average from participants and channels, thus small shifts between values will make amplitude values smaller as shown, *Figure 2a and b*.

## Time × time classification with time domain features

We adopted a time × time classification approach (temporal generalisation) on the time domain features. We first smoothed the EEG signals using 100 ms window such that every time point was replaced with the average of the 50 ms of both sides around it. We used a searchlight analysis to locate channels of interest. This used data from 12 participants who had performed the same SRTT task as those in the current study, but who had not undergone REM TMR and were therefore not included in our current dataset. Our classification used the smoothed amplitudes as features and LDA classifiers with fivefold cross-validation. In our searchlight analysis, the classifier was trained for

every channel and time points were used as features. Thus, a classification outcome was obtained for every channel as shown in *Figure 2—figure supplement 4*. In the time × time classification, EEG signals of sleep and wake are organised as 3D tensors where the dimensions were: trials × channels × time points. We then trained an LDA classifier at every time point from sleep (trials × channels), with the values from all the different channels as our features. The classifier from each time point in sleep was applied to all time points from wake to obtain one row of classification results in the time × time classification plot (e.g., *Figure 2—figure supplement 5*). The process was then repeated for all time points after a sound cue in sleep (trial length was: 1.5 s in sleep and 1.1 s in wake). Classifiers were first trained on sleep data from a participant and tested on wake data from that same participant, then the AUC values from different participants were averaged. In sleep, only sounds from the cued sequence were used. Two of the sounds (1=male face, 2=lamp) were associated with left hand and thus their trials were aggregated to form the left-hand class. The other two sounds (3=female face, 4=water tap) were associated with the right hand so their trials were aggregated to form the right-hand class. When testing the classifier on wake data, we were interested in the differential activation pattern for left- and right-handed motor imagery. We therefore included motor imagery data from both cued and un-cued sequences and from both pre-and post-sleep imagery sessions. There was an issue with the recorded data of pre-sleep imagery for one participant, thus, for that participant only the post-sleep imagery was used.

We delivered TMR cues as long as participants were in stable REM. This has the advantage of allowing us to train our classification models with many trials but in the meantime, we needed to make sure that our TMR cues were processed by the brain. We knew the pattern of brain response elicited after TMR in both experimental and adaptation nights as shown in *Figure 2a and b*, and used this to assess the success of TMR in eliciting a response. We thus incorporated fidelity of TMR response to this pattern in our model. We did this by measuring the average theta power at [4 8]Hz in the first second after TMR. We then averaged these values across channels and time, giving one value for each sleep trial. We then robustly normalised the values by subtracting the median and dividing by the inter-quartile range for each value. Next, we rescaled the values to have the range [0 1] to act as weights for trials when performing classification. Values that were higher than the 90th percentile were set to 1 and those lower than the 10th percentile were set to 0, with normalisation and flooring and capping weights reflect the fidelity of each TMR cue. We then gave those weights for the classifiers. This process was done for both the experimental and the adaptation nights. This provides a new way of informing the classifier of the fidelity of each TMR cue. We also report the number of trials for each condition in *Supplementary file 1*. We used the AUC as the performance measure in our classification. Analyses were done in Matlab using FieldTrip (*Oostenveld et al., 2011*), MVPA-Light (*Treder, 2020*), Lively Vectors (LV) (*Abdellahi, 2022*), and customised scripts using *Matlab, 2018*. The clustering window used for cluster-based permutation was the whole length of trials (i.e., the whole time × time classification duration) which gave a stringent test. In other words, we did not limit the clustering window to a specific time window.

## Behavioural analysis

The most robust measure of performance on the SRTT comes from the best performance blocks in each session (*Cousins et al., 2014*), we therefore took the average of the best four blocks before and after sleep to calculate performance. The SRTT measure which consolidates most strongly with sleep (*Spencer et al., 2006*) and with TMR (*Cousins et al., 2014*; *Cousins et al., 2016*) is SSS. This was calculated by subtracting reaction times on sequenced blocks from reaction times on the associated random blocks: SSS = (mean of random blocks – mean of best four blocks). Behavioural improvement across sleep was then calculated by: (SSS post-sleep – SSS pre-sleep)/SSS pre-sleep. In order to determine whether there are differences in this measure as a result of TMR, we calculated the relative difference in overnight improvement for cued and un-cued sequences as follows: relative sequence improvement (cued vs un-cued) = (improvement across sleep of the cued sequence – improvement across sleep of the un-cued sequence)/improvement across sleep of the un-cued sequence. When we correlated the relative cued vs. un-cued sequence improvement with classification performance, we calculated the latter as the mean of classification AUC values from each participant inside the cluster shown in *Figure 2c*.

## Correcting for multiple comparisons

Multiple comparisons correction was done using MVPA-Light toolbox in Matlab (*Treder, 2020*) and customised scripts. In cluster-based permutation, we had a 2D time × time classification from each participant and each condition (experimental and adaptation). Each 2D classification outcome in the adaptation was subtracted from its corresponding classification outcome in the experimental night. Afterwards, each point of that 2D difference was tested among participants to see whether it significantly differs from 0 or not. This gave a 2D z-statistic values at each point. Then, the z-statistic values of the significant contiguous points were added together to get the observed cluster statistic. Afterwards, permutations were done where 2D classification outcomes were randomly shuffled between conditions and the process of taking the difference, computing z-values to see significant points, and adding the z-values of the significant points is repeated. This was one shuffling iteration, we repeated this 10,000 times and constructed a distribution of those permutation results and tested whether our observed statistic of the actual data was significant. Cluster-based permutation with sample-specific threshold of 0.05 was used. Permutation test threshold for clusters was 0.05.

## Acknowledgements

We would like to thank Martyna Rakowska and Lorena Santamaria and all members of our group NaPs for the useful comments. This work was funded by the ERC grant SolutionSleep, 681607, to PAL.

## Additional information

### Funding

| Funder | Grant reference number | Author |
|---|---|---|
| European Research Council | 681607 | Penelope A Lewis |

The funders had no role in study design, data collection and interpretation, or the decision to submit the work for publication.

### Author contributions

Mahmoud EA Abdellahi, Software, Formal analysis, Validation, Investigation, Visualization, Methodology, Writing – original draft, Writing – review and editing; Anne CM Koopman, Conceptualization, Data curation, Investigation, Writing – review and editing; Matthias S Treder, Investigation, Methodology, Writing – review and editing; Penelope A Lewis, Conceptualization, Supervision, Validation, Investigation, Methodology, Project administration, Writing – review and editing

### Author ORCIDs

Mahmoud EA Abdellahi (ORCID) http://orcid.org/0000-0002-7765-9028
Penelope A Lewis (ORCID) http://orcid.org/0000-0003-1793-3520

### Ethics

This study was approved by the School of Psychology, Cardiff University Research Ethics Committee, and all participants gave written informed consents. Information of the participants are anonymised. Reference: EC.16.11.08.4772RA2. Risk Assessment: 1479917576_1583.

### Decision letter and Author response

Decision letter https://doi.org/10.7554/eLife.84324.sa1
Author response https://doi.org/10.7554/eLife.84324.sa2

## Additional files

### Supplementary files

- Supplementary file 1. The number of stimulations for left and right hand in sleep.
- MDAR checklist

## Data availability

All relevant data generated or analysed are available along with Matlab scripts. Data are available at the Open Science Framework (OSF): https://osf.io/wmyae/; https://osf.io/fq7v5/.

The following datasets were generated:

| Author(s) | Year | Dataset title | Dataset URL | Database and Identifier |
|---|---|---|---|---|
| Abdellahi MEA | 2022 | REM sleep TMR in human REM sleep elicits detectable reactivation with TMR | https://osf.io/wmyae/ | Open Science Framework, wmyae |
| Abdellahi MEA | 2022 | TMR in human REM sleep elicits detectable reactivation part2 | https://osf.io/fq7v5/ | Open Science Framework, fq7v5 |

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
