## [Editor Report]

This valuable work in human subjects reports that sounds that were associated with specific memories during waking behaviors can trigger the reactivation of these memory representations during REM sleep. Convincing evidence is provided to support the conclusions. The work expands our understanding of memory processing during sleep.

---

## [Decision Letter]

**Decision letter after peer review:**

Thank you for submitting your article "Targeted memory reactivation in human REM sleep elicits detectable reactivation" for consideration by *eLife*. Your article has been reviewed by 3 peer reviewers, and the evaluation has been overseen by Laura Colgin as the Senior Editor. The following individual involved in the review of your submission has agreed to reveal their identity: Kenneth A Norman (Reviewer #3).

Essential revisions:

1) Reviewers agreed that there is a lack of clarity in the Methods as currently written which make it difficult to evaluate the results (as one example, what features go into the classifier?). The Methods will need to be revised to provide clarification before a decision can be made about whether the claims in this manuscript are rigorously supported by the results. Please see individual reviews below for details.

2) Some claims are not well supported by the presented results. Additional analyses are suggested that may strengthen the evidence for some of the claims. Claims with insufficient support (e.g., claims related to theta, claims related to dilation/compression, etc.) should be removed from the paper unless additional analyses/results can be included to provide sufficient support. Please see individual reviews below for details.

3) The justification for excluding trials (e.g., discarding low variance trials and discarding trials to match the number of clear trials across participants) is unsatisfactory. Please see individual reviews below for details.

*Reviewer #1 (Recommendations for the authors):*

My specific suggestions are as follows:

I suggest including a section in the methods explaining how TMR sounds were presented during REM sleep.

Please describe the methods for the searchlight analysis used to select channels of interest and a region of interest.

It would be helpful to have more details about how the LDA classifier was trained on the sleep data. It is my understanding that each sleep trial began with a TMR cue (a sequence of tones), and that both the cued and uncued sequences were associated with both left and right-handed button presses (each sequence contains 1,2,3, and 4). It is unclear to me how the classifier is then trained to discriminate between left vs. right hand.

It would also be helpful to have more details about testing on the imagery data, including more details about how the imagery data were prepared and preprocessed. Was the classifier tested on the imagery data from both pre- and post-sleep, from both cued and uncued sequence trials?

Regarding the procedure of rejecting trials with low variance ('Because TMR will not be effective with all trials, we also rejected trials with a low variance that do not differ from their mean across time since they are unlikely to contain a response.') – my reading of this is that "a response" refers to reactivation. If that is the case, then trials from the adaptation night would not contain "a response". Or perhaps a response more broadly refers to any event-related response to the sound cue. To alleviate any concerns related to this, it might be helpful to include some more descriptive information about how many trials are included in various conditions/analyses.

*Reviewer #3 (Recommendations for the authors):*

One recommendation is for the authors to clarify the methods. In particular:

The description of the "searchlight" procedure that was used for feature selection lacked detail. I couldn't figure out what the authors did from what was written in the paper. What features were included in each searchlight? Did the searchlights encompass multiple electrodes? What frequencies were included for each electrode? or did the authors use raw power? Also, I don't understand the statement on line 354 that "this was done on different participants who performed the same task to avoid circularity". Does this mean that feature selection was done separately for each participant, using a nested cross-validation procedure? or by "different participants" do you mean "participants who were not included in the main sample for this study"? Much more detail is needed here.

Is the main classification analysis done within participants (i.e., train on one participant's sleep data, test on the same participant's wake data) or across participants? I am assuming the former, but I tried looking this up and I could not find it (apologies if I missed it). Relatedly, for the statistics, is AUC first computed within each subject and then averaged across subjects, or is it computed differently?

Line 375 mentions a PCA step in the time dilation analysis, but it is unclear what the features are that are being fed into the PCA (are the feature vectors of dimensionality n_frequencies x n_electrodes x n_timepoints, or are they something else, and is the PCA looking at variance across trials, or across time points within a trial, or something else).

More information on the cluster-based permutation (i.e., what was permuted?) would be useful.

My second recommendation is for the authors to revisit the issues mentioned in the public review regarding theta-mediation, time dilation/compression, and sequence-specificity of the classification-behavior relationship. If the authors can not strengthen their claims, they should step back from making these claims.

Regarding the analyses looking at theta: The authors' median split approach is not the most statistically powerful way to address this question (see, e.g., papers by Gary McClelland about median splits). Using a more continuous regression approach might yield better results.

Regarding the analyses looking at dilation/compression: Here, the authors' main task is to show that their results necessitate a dilation/compression explanation, as opposed to simply being due to autocorrelation in feature patterns over time.

Regarding the analyses looking at the "sequence-specificity" of the classifier-behavior relationship: To assess sequence-specificity, the authors could run a bootstrap where, for each resampling of subjects, they compute the difference between the fisher z of the cued sequence and the fisher z of the uncued sequence.

I was intrigued by the authors' use of training on sleep data and testing on wake data. The justification for this (that features that are reactivated during sleep may be a subset of features that are reactivated during wake) is a sensible hypothesis but conjectural. If this hypothesis is correct, it implies that results should be worse if classifiers are trained on wake and tested during sleep. I don't think it's necessary to do this, but if the authors tried the reverse approach it would be useful to see the results (if wake-trained classifiers are indeed worse than sleep-trained classifiers, this would provide some converging support for the authors' claim that reactivated features during sleep are a subset of features activated during wake). It might also be useful to see the results if the authors train on sleep and test on sleep.

For the analyses looking at whether theta power is used for classification, the authors perform a "negative control" where they include theta-band oscillations (but filter out other bands) and show that reactivation goes away. They may also want to include a "positive control" where they filter out theta-band oscillations and show that reactivation persists.

The fact that objects and faces were associated with the motor responses (crossed with the left and right responses) suggests to me that the authors may have considered looking at object/face classification in addition to the left/right classification. If the authors did in fact perform other analyses (that yielded null results) I think it would add to the value of the paper to report them here.

The authors match the number of "clean trials" across participants, using the smallest number that was obtained across all participants (366) – this seems like it is unnecessarily discarding useful data. If the AUC is computed within each subject and then averaged across subjects, then each subject will "count the same" regardless of how many trials are included.

Lines 300-302: why did you include these four blocks that contained random sequences?

How do the authors reconcile the overall lack of benefit from REM TMR with the finding that, across participants, the degree of reactivation evoked by REM TMR cues correlates with behavioural change? The explanation on lines 244-246 ("This absence of behavioural improvement could be due to individual differences in the way REM processes the task, potentially relating to an interaction between the extent or style of learning and REM processes") was not helpful to me in reconciling these two points – the authors might want to further unpack their argument here to make it clearer.

Phrase: "wake-like memory reactivation" – I am not sure what this phrase means.

---

## [Author Response]

Essential revisions:1) Reviewers agreed that there is a lack of clarity in the Methods as currently written which make it difficult to evaluate the results (as one example, what features go into the classifier?). The Methods will need to be revised to provide clarification before a decision can be made about whether the claims in this manuscript are rigorously supported by the results. Please see individual reviews below for details.

We have rewritten the methods so that it is clearer and have considered all the points that reviewers raised and conducted further analyses as requested.

2) Some claims are not well supported by the presented results. Additional analyses are suggested that may strengthen the evidence for some of the claims. Claims with insufficient support (e.g., claims related to theta, claims related to dilation/compression, etc.) should be removed from the paper unless additional analyses/results can be included to provide sufficient support. Please see individual reviews below for details.

We thank the reviewers for this comment. We have removed the dilation/compression analysis from the manuscript, and altered the theta analysis.

3) The justification for excluding trials (e.g., discarding low variance trials and discarding trials to match the number of clear trials across participants) is unsatisfactory. Please see individual reviews below for details.

We have now altered the analysis to avoid exclusion of trials in this way. The new analysis showed the same effect which is reassuring. We have updated the manuscript accordingly.

Reviewer #1 (Recommendations for the authors):My specific suggestions are as follows:I suggest including a section in the methods explaining how TMR sounds were presented during REM sleep.

Thanks for this comment. We have now included it as follows:

“The delivery of TMR cues during REM sleep

Cues were delivered to participants during stable REM sleep which was identified using standard American Academy of Sleep Medicine (AASM) criteria (Richard et al., 2015). Tones were presented in the correct order for our sequence, with a pause of 1500ms in between tones in a sequence, and a 20-second break at the end of each sequence. If participants showed any signs of arousal, or left REM sleep, cueing was paused immediately. Cueing then continued from where it had left off when they returned to stable REM sleep again.”

Please describe the methods for the searchlight analysis used to select channels of interest and a region of interest.

We have now updated the methods section and included more details about this in the section on time x time classification with time domain features:

“We used a searchlight analysis to locate channels of interest. This used data from 12 participants who had performed the same SRTT task as those in the current study, but who had not undergone REM TMR and were therefore not included in our current dataset. Our classification used the smoothed amplitudes as features and LDA classifiers with five-fold cross-validation. In our searchlight analysis, the classifier was trained for every channel and time points were used as features. Thus, a classification outcome was obtained for every channel as shown in (Figure 2—figure supplement 4).”

It would be helpful to have more details about how the LDA classifier was trained on the sleep data. It is my understanding that each sleep trial began with a TMR cue (a sequence of tones), and that both the cued and uncued sequences were associated with both left and right-handed button presses (each sequence contains 1,2,3, and 4). It is unclear to me how the classifier is then trained to discriminate between left vs. right hand.

Thanks for raising this point. This is very important, and we apologise that we were not clear enough in describing this before. Each sequence was associated with 4 tones (one for each finger), and the cued sequence used four completely different tones from the un-cued sequence (counterbalanced across participants). Each sleep trial began with just one TMR tone – e.g. the sound associated with just one finger used in the cued sequence. Our LDA classifiers were trained at every time point in the sleep data after each TMR tone onset, and a classifier model was built. Classes 1 and 2 were aggregated to represent the left-hand class, similarly, classes 3 and 4 were aggregated to represent the right-hand class. This model was then tested on each time point in wake after the presentation of each sound. We have edited this part in the methods to make the point clearer.

“In sleep, only sounds from the cued sequence were used. Two of the sounds (1 = male face, 2 = lamp) were associated with left hand and thus their trials were aggregated to form the left-hand class.”

It would also be helpful to have more details about testing on the imagery data, including more details about how the imagery data were prepared and preprocessed. Was the classifier tested on the imagery data from both pre- and post-sleep, from both cued and uncued sequence trials?

We were interested in the activation pattern of right- and left-hand motor imagery which should not be dependent on the tone, thus, the classifier was tested on imagery data from the cued and un-cued sequences and we included all data from pre- and post-sleep motor imagery. Data were collected at 100 Hz sampling rate and were band-pass filtered in the range [0.1 50] Hz, then segmented into trials of 1100ms. We have now included more details about this in the text:

“When testing the classifier on wake data, we were interested in the differential activation pattern for left- and right-handed motor imagery. We therefore included motor imagery data from both cued and un-cued sequences and from both pre-and post-sleep imagery sessions. There was an issue with the recorded data of pre-sleep imagery for one participant, thus, for that participant only the post-sleep imagery was used.”

Reviewer #3 (Recommendations for the authors):One recommendation is for the authors to clarify the methods. In particular:The description of the "searchlight" procedure that was used for feature selection lacked detail. I couldn't figure out what the authors did from what was written in the paper. What features were included in each searchlight? Did the searchlights encompass multiple electrodes? What frequencies were included for each electrode? or did the authors use raw power? Also, I don't understand the statement on line 354 that "this was done on different participants who performed the same task to avoid circularity". Does this mean that feature selection was done separately for each participant, using a nested cross-validation procedure? or by "different participants" do you mean "participants who were not included in the main sample for this study"? Much more detail is needed here.

Thanks for this comment and sorry that we were not clear. We have now updated the methods section and included more details about this in the section on time x time classification with time domain features:

“We used a searchlight analysis to locate channels of interest. This used data from 12 participants who had performed the same SRTT task as those in the current study, but who had not undergone REM TMR and were therefore not included in our current dataset. Our classification used the smoothed amplitudes as features and LDA classifiers with five-fold cross-validation. In our searchlight analysis, the classifier was trained for every channel and time points were used as features. Thus, a classification outcome was obtained for every channel as shown in (Figure 2—figure supplement 4).”

Is the main classification analysis done within participants (i.e., train on one participant's sleep data, test on the same participant's wake data) or across participants? I am assuming the former, but I tried looking this up and I could not find it (apologies if I missed it). Relatedly, for the statistics, is AUC first computed within each subject and then averaged across subjects, or is it computed differently?

Thanks for this point. Yes, the classification was done by training on one participant and testing on the same participant, we now added this in the text. Yes, the AUC was calculated for each participant and then averaged across participants, we have included this now in the text as well.

“Classifiers were first trained on sleep data from a participant and tested on wake data from that same participant, then the AUC values from different participants were averaged.”

Line 375 mentions a PCA step in the time dilation analysis, but it is unclear what the features are that are being fed into the PCA (are the feature vectors of dimensionality n_frequencies x n_electrodes x n_timepoints, or are they something else, and is the PCA looking at variance across trials, or across time points within a trial, or something else).

In response to your comments and those of other reviewers we have now removed the analysis of temporal compression and dilation. However, we have tried to answer this query:

“raw data was smoothed and used as time domain features. The data was then organized as trials x channels x timepoints then we segmented each trial in time based on the compression factor we are using. For instance, if we test if sleep is 2x faster than wake we look at the trial lengths in wake which was 1.1 sec. and we take half of this value which is 0.55 sec. we then take different window in time from sleep data such that each sleep trial will have multiple smaller segments each of 0.55 sec., we then add those segments as new trials and label them with the respective trial label. Afterward, we resize those segments temporally to match the length of wake trial. We now reshape our data from trials x channels x timepoints to trials x channels_timepoints so we aggregate channels and timepoints into one dimension. We then feed this to PCA to reduce the dimensionality of channels_timepoints into principal components. We then feed the resultant features to a LDA classifier for classification. This whole process is repeated for every scaling factor and it is done within participant in the same fashion the main classification was done and the error bars were the standard errors and we tested the results from the experimental night to those of that adaptation night.”

More information on the cluster-based permutation (i.e., what was permuted?) would be useful.

We have now added the details of cluster-based permutation:

“In cluster-based permutation, we had a 2d time x time classification from each participant and each condition (experimental and adaptation). Each 2d classification outcome in the adaptation was subtracted from its corresponding classification outcome in the experimental night. Afterwards, each point of that 2d difference was tested among participants to see whether it significantly differs from 0 or not. This gave a 2d z-statistic values at each point. Then the z-statistic values of the significant contiguous points were added together to get the observed cluster statistic. Afterwards, permutations were done where 2d classification outcomes were randomly shuffled between conditions and the process of taking the difference, computing z-values to see significant points, and adding the z-values of the significant points is repeated. This was one shuffling iteration, we repeated this 10,000 times and constructed a distribution of those permutation results and tested whether our observed statistic of the actual data was significant.”

My second recommendation is for the authors to revisit the issues mentioned in the public review regarding theta-mediation, time dilation/compression, and sequence-specificity of the classification-behavior relationship. If the authors can not strengthen their claims, they should step back from making these claims.Regarding the analyses looking at theta: The authors' median split approach is not the most statistically powerful way to address this question (see, e.g., papers by Gary McClelland about median splits). Using a more continuous regression approach might yield better results.

Thanks for the comment. In response to your comments and those of other reviewers we have now altered this analysis and no longer split the trials into high and low theta.

Regarding the analyses looking at dilation/compression: Here, the authors' main task is to show that their results necessitate a dilation/compression explanation, as opposed to simply being due to autocorrelation in feature patterns over time.

Thank you. We have removed the analysis of temporal compression and dilation from the manuscript, so our answer is no longer relevant to the review process. However, we provide some details about this analysis. Raw data was smoothed and used as time domain features. The data was then organized as trials x channels x timepoints then we segmented each trial in time based on the compression factor we are using. For instance, if we test if sleep is 2x faster than wake we look at the trial lengths in wake which was 1.1 sec. and we take half of this value which is 0.55 sec. we then take a different window in time from sleep data such that each sleep trial will have multiple smaller segments each of 0.55 sec., we then add those segments as new trials and label them with the respective trial label. Afterwards, we resize those segments temporally to match the length of wake trials. We now reshape our data from trials x channels x timepoints to trials x channels_timepoints so we aggregate channels and timepoints into one dimension. We then feed this to PCA to reduce the dimensionality of channels_timepoints into principal components. We then feed the resultant features to a LDA classifier for classification. This whole process is repeated for every scaling factor and it is done within participant in the same fashion the main classification was done and the error bars were the standard errors. We compared the results from the experimental night to those of the adaptation night.

Regarding the analyses looking at the "sequence-specificity" of the classifier-behavior relationship: To assess sequence-specificity, the authors could run a bootstrap where, for each resampling of subjects, they compute the difference between the fisher z of the cued sequence and the fisher z of the uncued sequence.

Thank you for the suggestion. We have now considered comments regarding sequence specificity, and have now examined sequence specific improvement by subtracting the improvement of the un-cued sequence from improvement on the cued sequence and then normalising the result by the improvement of the un-cued sequence. The resulting value, which we term ‘cued sequence improvement’ correlated significantly with classification performance (n = 14, r = 0.56, p = 0.04).

We have updated the text as follows:

“We therefore set out to determine whether there was a relationship between the extent to which we could classify reactivation and overnight improvement on the cued sequence. This revealed a positive correlation (n = 14, r = 0.56, p = 0.04), Figure 3b.”

I was intrigued by the authors' use of training on sleep data and testing on wake data. The justification for this (that features that are reactivated during sleep may be a subset of features that are reactivated during wake) is a sensible hypothesis but conjectural. If this hypothesis is correct, it implies that results should be worse if classifiers are trained on wake and tested during sleep. I don't think it's necessary to do this, but if the authors tried the reverse approach it would be useful to see the results (if wake-trained classifiers are indeed worse than sleep-trained classifiers, this would provide some converging support for the authors' claim that reactivated features during sleep are a subset of features activated during wake). It might also be useful to see the results if the authors train on sleep and test on sleep.

We like the suggestion of looking at the reverse approach, and therefore decided to conduct this extra analysis. This indeed showed that the classification performance is reduced when wake was used to train the classifiers as compared to when sleep was used. Figure 2-figure supplement 3 shows the classification results for (a) training the classifiers with wake and testing with sleep (b) training the classifiers with sleep and testing with wake. The difference in the AUC values is clear. To make sure that this is consistent among participants and not due to averaging the classification performance, we compared each of the classification outcomes to their corresponding classification of the adaptation night which gave two 2d zscores and in (c) we show the difference between the two zscores, when the zscore of the wake trained is subtracted from sleep trained approach. The difference of zscores is positive and higher than 2 around 1sec. which was the time of the cluster that gave significant effect in the original analysis. This shows that training classifiers with sleep enabled the classifiers to adapt to sleep data.

For the analyses looking at whether theta power is used for classification, the authors perform a "negative control" where they include theta-band oscillations (but filter out other bands) and show that reactivation goes away. They may also want to include a "positive control" where they filter out theta-band oscillations and show that reactivation persists.

Thanks for the suggestion. We have conducted this analysis and now include the results of the positive control where we filter out theta-band [4 8]Hz oscillations. This shows that reactivation evidence was significant against both chance and the adaptation night when non-theta band frequencies are included:

“For completeness, we also conducted a positive control in which we filtered-out theta band and performed our classification analysis. This showed a similar significant pattern as we had observed using the full spectrum without filtering, e.g. as in Figure 2c, d (n=14, p=0.007, corrected with cluster-based permutation vs. chance level) and (n=14, p=0.009, corrected with cluster-based permutation vs. the adaptation night, as shown in Figure 2—figure supplement 2).”

The z-statistic values and the classification AUC are now included as Figure 2—figure supplement 2.

The fact that objects and faces were associated with the motor responses (crossed with the left and right responses) suggests to me that the authors may have considered looking at object/face classification in addition to the left/right classification. If the authors did in fact perform other analyses (that yielded null results) I think it would add to the value of the paper to report them here.

Thanks for this comment. We had a priori-hypothesis about classification of left and right hands as we found higher classification rates with right- and left-hand previously with our classification of this task (Abdellahi et al., 2023). In the future we may try to perform classification of objects and faces with different features or perform a multi-class classification of the four classes with different features and test the reactivation of individual items, but that is outside the scope of this paper.

The authors match the number of "clean trials" across participants, using the smallest number that was obtained across all participants (366) – this seems like it is unnecessarily discarding useful data. If the AUC is computed within each subject and then averaged across subjects, then each subject will "count the same" regardless of how many trials are included.

Thanks for this. We have now changed this part of our analysis and no longer fix the number of trials at 366. We now also report the number of trials in supplementary file 1.

Lines 300-302: why did you include these four blocks that contained random sequences?

The random sequence was used to help us calculate the Sequence Specific Skill (SSS), the SRTT measure which consolidates most strongly with sleep (Spencer et al., 2006) and with TMR (Cousins et al., 2014, 2016). Specifically, this is a measure of how each fixed sequence improved in relation to the associated random sequence, showing how much of the improvement was sequence specific (Cousins et al., 2014, 2016; Rakowska et al., 2021). This is achieved by subtracting the reaction times of the fixed sequence from the random sequence, thus if the reaction times of the fixed sequence improves a lot across sleep, and those of the random sequence do not then there is a big difference which, will show up as higher SSS (Koopman et al., 2020; Rakowska et al., 2021).

How do the authors reconcile the overall lack of benefit from REM TMR with the finding that, across participants, the degree of reactivation evoked by REM TMR cues correlates with behavioural change? The explanation on lines 244-246 ("This absence of behavioural improvement could be due to individual differences in the way REM processes the task, potentially relating to an interaction between the extent or style of learning and REM processes") was not helpful to me in reconciling these two points – the authors might want to further unpack their argument here to make it clearer.

Thank you, we now discuss this as:

“We are at a very early phase in understanding what TMR does in REM sleep, however we do know that the connection between hippocampus and neocortex is inhibited by the high levels of Acetylcholine that are present in REM (Hasselmo, 1999). This means that the reactivation which we observe in the cortex is unlikely to be linked to corresponding hippocampal reactivation, so any consolidation which occurs as a result of this is also unlikely to be linked to the hippocampus. The SRTT is a sequencing task which relies heavily on the hippocampus, and our primary behavioural measure (Sequence Specific Skill) specifically examines the sequencing element of the task. Our own neuroimaging work has shown that TMR in non-REM sleep leads to extensive plasticity in the medial temporal lobe (Cousins et al., 2016). However, if TMR in REM sleep has no impact on the hippocampus then it is quite possible that it elicits cortical reactivation and leads to cortical plasticity but provides no measurable benefit to Sequence Specific Skill. Alternatively, because we only measured behavioural improvement right after sleep it is possible that we may have missed behavioural improvements that would have emerged several days later, as we know can occur in this task (Rakowska et al., 2021).”

Phrase: "wake-like memory reactivation" – I am not sure what this phrase means.

We have now changed this phrase:

Abstract:

“These findings provide the first evidence for memory reactivation in human REM sleep after TMR that is directly related to brain activity during wakeful task performance.”

Introduction:

“Nevertheless, the reinstatement of EEG activity in REM that directly relates to EEG activity during a task in wake has yet to be demonstrated in humans.”

Discussion:

“In this paper we demonstrated that memory reactivation after TMR cues in human REM sleep can be decoded using EEG classifiers. Such reactivation appears to be most prominent about one second after the sound cue onset.”

References

Abdellahi, M. E. A., Koopman, A. C. M., Treder, M. S., and Lewis, P. A. (2023). Targeting targeted memory reactivation: characteristics of cued reactivation in sleep. *NeuroImage*, *266*, 2021.12.09.471945. https://www.sciencedirect.com/science/article/pii/S1053811922009417?via%3Dihub

Cairney, S. A., Guttesen, A. á. V., El Marj, N., and Staresina, B. P. (2018). Memory Consolidation Is Linked to Spindle-Mediated Information Processing during Sleep. *Current Biology*, *28*(6), 948-954.e4. https://doi.org/10.1016/j.cub.2018.01.087

Cousins, J. N., El-Deredy, W., Parkes, L. M., Hennies, N., and Lewis, P. A. (2014). Cued memory reactivation during slow-wave sleep promotes explicit knowledge of a motor sequence. *Journal of Neuroscience*, *34*(48), 15870–15876. https://doi.org/10.1523/JNEUROSCI.1011-14.2014

Cousins, J. N., El-Deredy, W., Parkes, L. M., Hennies, N., and Lewis, P. A. (2016). Cued Reactivation of Motor Learning during Sleep Leads to Overnight Changes in Functional Brain Activity and Connectivity. *PLoS Biology*, *14*(5), e1002451. https://doi.org/10.1371/journal.pbio.1002451

Diekelmann, S., and Born, J. (2010). The memory function of sleep. In *Nature Reviews Neuroscience*. https://doi.org/10.1038/nrn2762

Hasselmo, M. E. (1999). Neuromodulation: Acetylcholine and memory consolidation. In *Trends in Cognitive Sciences*. https://doi.org/10.1016/S1364-6613(99)01365-0

Koopman, A. C. M., Abdellahi, M. E. A., Belal, S., Rakowska, M., Metcalf, A., Śledziowska, M., Hunter, T., and Lewis, P. (2020). Targeted memory reactivation of a serial reaction time task in SWS, but not REM, preferentially benefits the non-dominant hand. *BioRxiv*, 2020.11.17.381913. https://doi.org/10.1101/2020.11.17.381913

Pereira, S. I. R., Beijamini, F., Vincenzi, R. A., and Louzada, F. M. (2015). Re-examining sleep’s effect on motor skills: How to access performance on the finger tapping task? *Sleep Science*. https://doi.org/10.1016/j.slsci.2015.01.001

Rakowska, M., Abdellahi, M. E. A., Bagrowska, P., Navarrete, M., and Lewis, P. A. (2021). Long term effects of cueing procedural memory reactivation during NREM sleep. *NeuroImage*. https://doi.org/10.1016/j.neuroimage.2021.118573

Richard, B., Rita, B., Charlene, E. G., Harding;, S. M., Lloyd;, R. M., Marcus, C. L., and Vaughn, B. V. (2015). The AASM Manual for the Scoring of Sleep and Associated Events. *American Academy of Sleep Medicine*.

Schönauer, M., Alizadeh, S., Jamalabadi, H., Abraham, A., Pawlizki, A., and Gais, S. (2017). Decoding material-specific memory reprocessing during sleep in humans. *Nature Communications*. https://doi.org/10.1038/ncomms15404

Schreiner, T., and Rasch, B. (2015). Boosting vocabulary learning by verbal cueing during sleep. *Cerebral Cortex*. https://doi.org/10.1093/cercor/bhu139

Spencer, R. M. C., Sunm, M., and Ivry, R. B. (2006). Sleep-Dependent Consolidation of Contextual Learning. *Current Biology*. https://doi.org/10.1016/j.cub.2006.03.094